# Multiple yet switchable hydrogen-bonded organic frameworks with white-light emission

Yadong Shi[1], Shuodong Wang[2], Wei Tao[1], Jingjing Guo[3], Sheng Xie[2], Yanglan Ding[1], Guoyong Xu[1], Cheng Chen[1], Xiaoyu Sun[4], Zengming Zhang [4], Zikai He [5], Peifa Wei [1,3]✉ & Ben Zhong Tang[6]✉

The development of new strategies to construct on-demand porous lattice frameworks from simple motifs is desirable. However, mitigating complexity while combing multiplicity and reversibility in the porous architectures is a challenging task. Herein, based on the synergy of dynamic intermolecular interactions and flexible molecular conformation of a simple cyano-modified tetraphenylethylene tecton, eleven kinetic-stable hydrogen-bonded organic frameworks (HOFs) with various shapes and two thermo-stable non-porous structures with rare perpendicular conformation are obtained. Multimode reversible structural transformations along with visible fluorescence output between porous and non-porous or between different porous forms is realized under different external stimuli. Furthermore, the collaborative of flexible framework and soft long-chain guests facilitate the relaxation from intrinsic blue emission to yellow emission in the excited state, which represents a strategy for generating white-light emission. The dynamic intermolecular interactions, facilitated by flexible molecular conformation and soft guests, diversifies the strategies of construction of versatile smart molecular frameworks.

---

[1] Institutes of Physical Science and Information Technology, Key Laboratory of Structure and Functional Regulation of Hybrid Materials of Ministry of Education, Anhui Graphene Engineering Laboratory, Anhui University, Hefei 230601, China. [2] State Key Laboratory of Chemo/Biosensing and Chemometrics, College of Chemistry and Chemical Engineering, Hunan University, Changsha 410082, China. [3] State Key Laboratory of Luminescent Materials and Devices, Guangdong Provincial Key Laboratory of Luminescence from Molecular Aggregates, South China University of Technology, Guangzhou 510640, China. [4] The Center for Physical Experiments, School of Physics Science, University of Science and Technology of China, Hefei 230026, China. [5] School of Science, Harbin Institute of Technology, Shenzhen 518055, China. [6] Shenzhen Institute of Aggregate Science and Technology, School of Science and Engineering, The Chinese University of Hong Kong, Shenzhen, Guangdong 518172, China. ✉email: pfwei@ahu.edu.cn; tangbenz@cuhk.edu.cn

Smart materials as one of the most advanced and promising interdisciplinary directions have witnessed their superiority in recent years. Especially those with switchable responsiveness to external stimuli such as light, electric or magnetic fields, temperature, and chemical compounds are attracting more attention due to their advantages in energy economy, efficiency improvement, and environmental friendliness[1–3]. From the view of molecular design, in order to endow the materials with switchability, it is generally necessary to decorate the molecule with corresponding responsive groups[4]. While whether the switchability at molecular level will eventually be transferred into macroscopic materials, such as solid material, is difficult to predict[5,6]. Considering most of the smart materials can only realize two-state switching[7,8], multiple yet controllable transition relies on much more tailored structural design, which will thus lead to tedious and time-consuming organic synthesis and purification[9]. Thus, mitigating complexity and combing multiplicity and reversibility in one simple molecule to construct multi-switchable materials remains a big challenge.

Self-assembly that is based on dynamic and tunable non-covalent interactions, such as hydrogen bonding[10–14], π–π[15,16], host-guest chemistry[17–20], electrostatics[21–24], etc., represents a potential and powerful strategy to cater to the aforementioned requirements. Thus, the question arises of how to effectively control interactions in the solid state. Allowing stimulus to interact with molecules in the bulk sample easily and deeply is the key to solve the problem[25,26]. Hydrogen-bonded organic frameworks (HOFs) with channels can provide a unique solution[27–29]. Compared with traditional extended frameworks, such as metal-organic frameworks constructed by strong coordination bonds[30] and covalent organic frameworks constructed by strong covalent bonds[31–34], the structures of HOFs constructed by weak interactions are much more flexible[35–37]. The tectons in most frameworks tend to maximize intermolecular interactions to form a collapsed thermodynamically stable structure[38–40], which is generally suppressed by guests in the channel[41,42]. However, if this tendency is used ingeniously when adsorption of various guests[43], the adaptive conformation of tectons and packing of HOFs can thus be tuned precisely, thereby delivering various HOFs.

Regarding to tectons' design, four factors should be considered: (i) its structure should be as simple as possible[44]; (ii) the topology of tectons should be geometrically favorable for the formation of framework[45–47]; (iii) molecular conformations should have sufficient flexibility which can be easily affected by external inputs[48–50]; (iv) for better responsiveness tracking and mechanisms understanding, it will be more perfect to encode visible signal output such as fluorescence, absorption, etc. into the transformation process[51–54]. An ideal, simple, and readily accessible tetraphenylethylene (TPE) molecule that can provide solid-state emission is thus selected[55–57]. The four-armed scaffolds with $C_2$ symmetry protruding radially produce directional interactions in space to direct pore formation. The molecular propeller conformation provides guarantee for the flexibility of the pore. Decorating functional groups on its four arms to further introduce multiple weak interactions such as hydrogen bonding, in combination with the inherent π-π stacking properties of the phenyl ring, will finally deliver multiple yet switchable HOFs.

Organic white-light-emitting materials have attracted a lot of research attention due to their important application prospects. So far, although there are many reports available on the construction of white-light-emitting systems, including nanostructures[58], supramolecules[59], small molecules,[60,61] hydrogels[62] and so on. However, developing simple strategies for preparing white-light-emitting materials is still highly desirable[63–66]. Frameworks with channels can accommodate guests of matching sizes through non-covalent bond interactions. The characteristics of the guests will have a non-negligible effect on the luminescence of the frameworks[49,55]. Thus, the combination of fluorescence and switchable HOFs induced by guests may provide a potential way for the construction of tunable luminescent materials, including white-light-emitting materials.

In this work, based on a simple cyano-modified TPE derivative named as 4CN, relying on its flexible molecular conformations, cooperative with tunable hydrogen bonding and π-π stacking modes, we construct eleven porous crystals and two nonporous crystals. The aggregation-induced emission (AIE)[67] character of the tecton endows the bulk crystals with variable emission ranges from blue to green. Through solvent-induced structural rearrangement combined with fluorescence, we achieve multiple visible switching between 13 different packings. More interestingly, considering the dynamic of the HOFs, as the flexible long-chain alkanes enter into the cavity, the synergy of the frameworks and the guests induce the assembly to relax in the excited state to generate a yellow emission band, which combines with the intrinsic blue emission of the framework to finally deliver a pure white-light emission (Fig. 1). Through the introduction of alkanes with different lengths, a variety of white-light emission systems are constructed, which proves the universality of this white-light emission strategy.

## Results

**Multiple non-white-light emitting HOFs.** Considering TPE is an archetypal AIE-active molecule, its derivative 4CN should also exhibit interesting optical properties. Its UV/vis spectrum in tetrahydrofuran shows absorption up to 327 nm (Supplementary Fig. 1). It was weakly emissive in acetone, but enhanced photoluminescence (PL) was observed with gradual addition of water to acetone solution or in other words, 4CN was AIE-active (Supplementary Fig. 2). The olefin stator is surrounded by benzonitrile rotors, and the restriction of intramolecular motion should account for its strong fluorescence in the aggregate state. The electrostatic potential (ESP) of 4CN was shown in Fig. 2c, and the negative electron areas were mainly located on the four cyano groups. Conversely, the positive electron areas were mainly

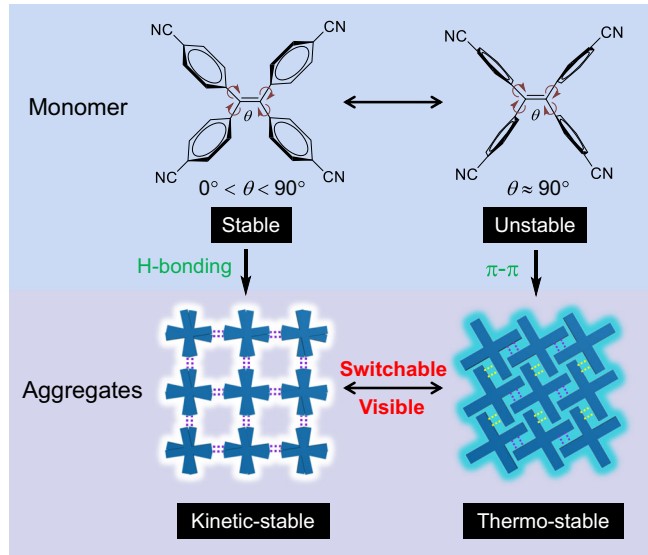

**Fig. 1 Tunable emission based on dynamic intermolecular interactions in switchable HOFs.** The purple and yellow dashed lines indicate hydrogen bonding and π-π interactions, respectively. 'θ' indicates the dihedral angles between four phenyl groups and the central ethenyl group.

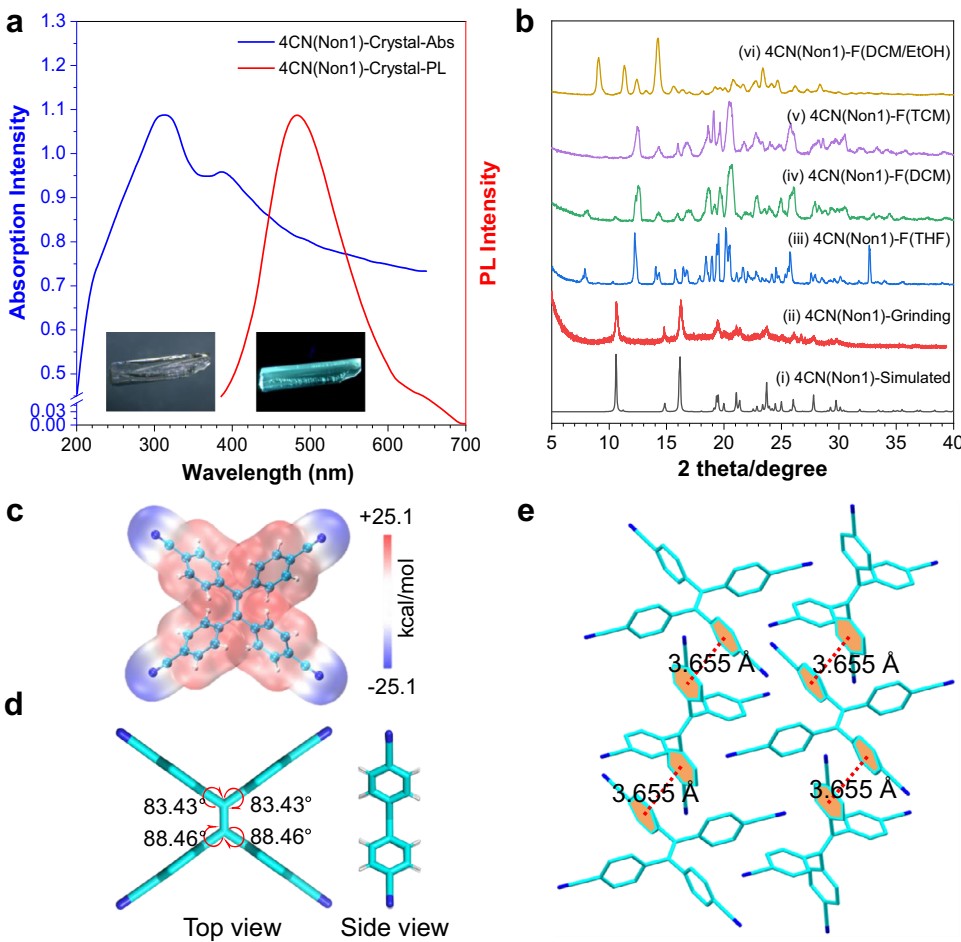

**Fig. 2 Investigation of single-crystal structures of 4CN(Non1). a** UV-Vis diffuse reflectance spectra (left) and PL spectra (right) of 4CN(Non1) crystals. $\lambda_{ex} = 365$ nm. Inset: the photograph of 4CN(Non1) crystals under daylight (left) and UV light (right). **b** The PXRD patterns of simulated 4CN(Non1) crystals, ground 4CN(Non1) crystals, fumed 4CN(Non1) crystals with different solvents. **c** ESP diagram of 4CN. **d** The dihedral angles between four phenyl groups and central ethenyl group of 4CN(Non1) crystals. **e** Crystal packing of 4CN(Non1) with labeled π–π stacking interactions.

located on the TPE. This promotes donor-acceptor interactions between adjacent TPE molecules, thereby facilitating π–π interactions of benzonitrile and hydrogen bonds formation during the crystallization process (Supplementary Fig. 10).

A colorless and transparent single crystal (Fig. 2a, blue line) named 4CN(Non1) with bright cyan emission at 482 nm (Fig. 2a, red line) was grown by slow vapor diffusion of cyclohexane into its tetrahydrofuran solution. 4CN(Non1) crystallized in monoclinic system and $C2/c$ space group (Supplementary Table 2) with solvent-free in the crystal lattice, in which 4CN molecules took a compact packing (Supplementary Fig. 8e). Thermogravimetric analysis (TGA) indicated 4CN(Non1) crystals exhibited excellent thermal stability with negligible weight loss even when the temperature was raised up to about 330 °C (Supplementary Fig. 25). Each 4CN molecule was connected with four neighboring molecules to form a supramolecular network through intermolecular π–π stacking interactions (3.655 Å, Fig. 2e and Supplementary Fig. 11d) and C–H···N hydrogen bonds (2.762 Å, 2.819 Å, 2.834 Å, 2.893 Å, 2.987 Å, 2.997 Å; Supplementary Fig. 11d), where nonradiative decay was suppressed due to close molecular packing. In terms of crystal form, for the symmetrical molecular structure without solvent, each benzonitrile wedge tilts almost perpendicularly to the connected olefin, adopting dihedral angles ($\theta$) between four phenyl groups and central ethenyl group of 83.43° and 88.46° (Fig. 2d; Supplementary Figs. 3 and 7c). It should be noted, this perpendicular conformation is rare in TPE derivatives[67]. To figure out the reason, we carried out

theoretical calculations with density functional theory at the b3lyp/6-31 G* level. The calculated dihedral angle of 4CN(Non1) changes to 48.40° in a gaseous environment, while in a solid environment keeps almost the same as those in the crystal state (Supplementary Figs. 4 and 5). This suggests intermolecular interactions should be the main driving force to stabilize the vertical phenyl groups in 4CN(Non1). However, the ground energy profile with different dihedral angles in a gaseous environment suggests that ~50° is the lowest energy point while 4CN with ~90° should be one of the unfavorable conformations (Supplementary Fig. 6). This inspires us to further study the stability of 4CN(Non1) under different stimuli.

For most cases, phase transition can be realized by grinding the crystal to amorphous state and then fuming with different solvents to generate various other crystalline states. So, the powder X-ray diffraction (PXRD) experiment of the ground 4CN(Non1) was firstly conducted (Fig. 2b(ii)). However, most of the resolvable peaks of the ground sample are consistent with the simulated PXRD pattern of 4CN(Non1), but the weaker intensity and broader peak shapes indicate the crystal structure of 4CN(Non1) can be influenced by an external stimulus, although the π–π stacking and hydrogen bonds are difficult to completely break by grinding. Then, we tried to fume the 4CN(Non1) crystals with different solvents directly (Fig. 2b(iii)–(vi)). To our surprise, the fumed samples show quite different PXRD, which means solvent fumigation can change the conformation of 4CN(Non1). Moreover, different solvents can induce the

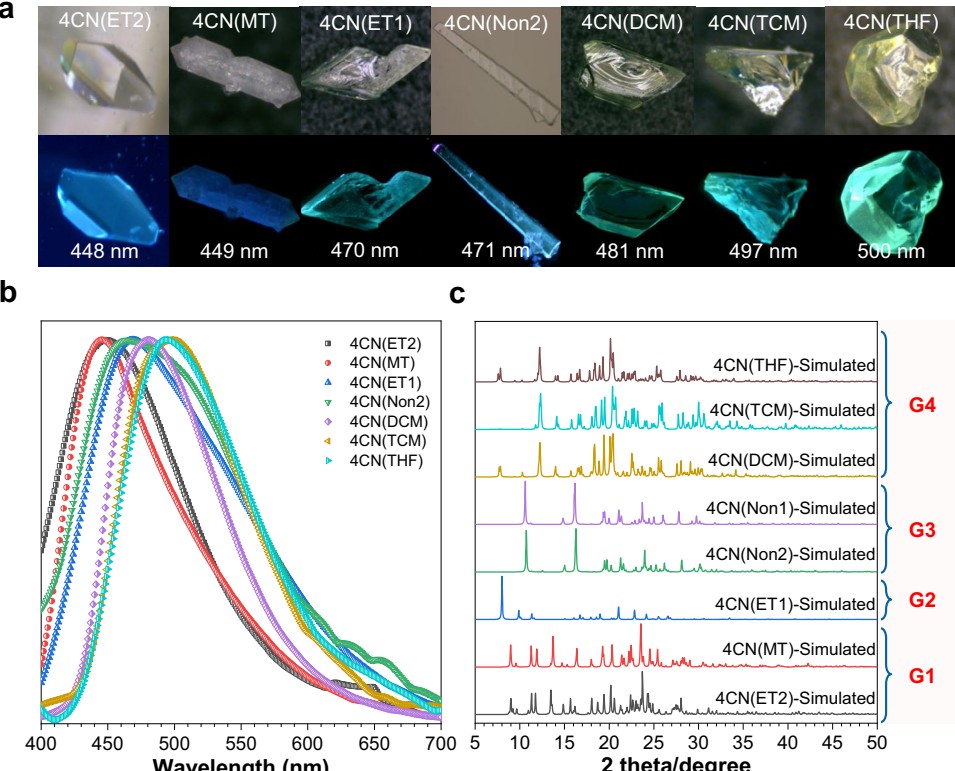

**Fig. 3 Multiple emissive crystals of 4CN. a** The photographs of different crystals of 4CN under daylight (first row) and UV light (second row). **b** Normalized fluorescent emission spectra of different crystals. **c** The corresponding simulated PXRD patterns of different crystals, that is, 4CN(ET2) and 4CN(MT) in group 1 (**G1**), 4CN(ET1) in group 2 (**G2**), 4CN(Non1) and 4CN(Non2) in group 3 (**G3**) or 4CN(THF), 4CN(DCM) and 4CN(TCM) in group 4 (**G4**).

generation of quite different packing. This suggests the intermolecular interactions and molecular conformation of 4CN can be exceptionally flexible in the solid-state.

In order to gain further insights, 4CN single crystals were cultured in different solvents. Surprisingly, we obtained another seven kinds of single crystals, those were 4CN(Non2) with no solvent, 4CN(THF) with tetrahydrofuran, 4CN(DCM) with dichloromethane, 4CN(TCM) with trichloromethane, 4CN(ET1)[68] and 4CN(ET2) with ethanol, and 4CN(MT) with methanol (Supplementary Table 2). Among them, the simulated PXRD of 4CN(THF), 4CN(DCM), 4CM(TCM), 4CN(ET2) crystals corresponds well to that of aforementioned fumed 4CN(Non1) crystals (Supplementary Fig. 12). These crystals emit rich and colorful emission, including blue, cyan, and blue-green (Fig. 3a). The maximum emission wavelength ($\lambda_{em}$) ranges from 448 to 500 nm (Fig. 3b). The quantum yield ($\Phi$) and lifetime ($\tau$) of all the crystals were measured (Supplementary Table 1 and Supplementary Fig. 22). According to the simulated PXRD (Fig. 3c), these eight crystals can be further categorized into four groups named **G1** (contains 4CN(ET2) and 4CN(MT)), **G2** (contains 4CN(ET1)), **G3** (contains 4CN(Non1) and 4CN(Non2)), and **G4** (contains 4CN(THF), 4CN(DCM), and 4CN(TCM)). The detailed crystal packing diagram viewed along different axes and the diagram with solvent-accessible void space suggests that **G1**, **G2**, and **G4** are all crystallized in a flexible-space-forming manner except for **G3** (Supplementary Fig. 8), that is to say, six different emissive HOFs are obtained.

The $\lambda_{em}$ and $\theta$ of these crystals are shown in Supplementary Fig. 7 and Supplementary Table 1. The $\lambda_{em}$ of the four groups generally conforms to the following sequence: **G1** < **G2** < **G3** < **G4**. The positive correlation between $\lambda_{em}$ and $\theta$ is anticipated as the large $\theta$

leads to the poor conjugation and thus blueshifts the $\lambda_{em}$. On the other hand, the cofacial π–π interaction will red shift the $\lambda_{em}$ as the interaction can stabilize the excited states. For **G1** and **G2**, the blue emission ranges from 440 to 470 nm can be attributed to the large dihedral angles ($\theta_{aver} \sim 55°$). The $\theta_{aver}$ in **G4** are around 45°, much smaller than that in **G1** and **G2**, which accounts for their good π-conjugation and relative red-shifted emission to 480–500 nm. A special case is **G3**, though the biggest dihedral angles ($\theta_{aver} > 70°$) will destroy the molecular π-conjugation, which should blue-shifted the emission, the strong intermolecular π–π interactions will force this trend to go in the opposite direction and finally leads to slightly red shift emission to cyan color (470–480 nm) (Supplementary Fig. 11c, d). In short, the colorful emission of crystallized 4CN can be ascribed to the synergy of flexible molecular conformation and dynamic intermolecular interactions.

**Multiple white light emitting HOFs.** It should be noted that bluish-green emissive 4CN(ET1) shows one-dimensional pores with a diameter of 7.4 Å and entrapped ethanol inside (Fig. 3b and Supplementary Fig. 14). The crystal structure with a total solvent-accessible void at 17.2% of its volume was estimated by *PLATON* analysis using a 1.2 Å probe[38,69]. Considering the flexibility of 4CN(ET1) framework, we wonder whether the introduction of much more flexible guests would influence the framework emission. Interestingly, five white-light emissive crystals, that is 4CN(Hex) (Supplementary Table 3), 4CN(Hep) (Supplementary Table 3), 4CN(Oct) (Supplementary Table 3), 4CN(Dec) (Supplementary Table 3), and 4CN(Dod) (Supplementary Table 3) with entrapped *n*-hexane, *n*-heptane, *n*-octane, *n*-decane, and *n*-dodecane are obtained (Fig. 4b and Supplementary Fig. 15). Due to the similar

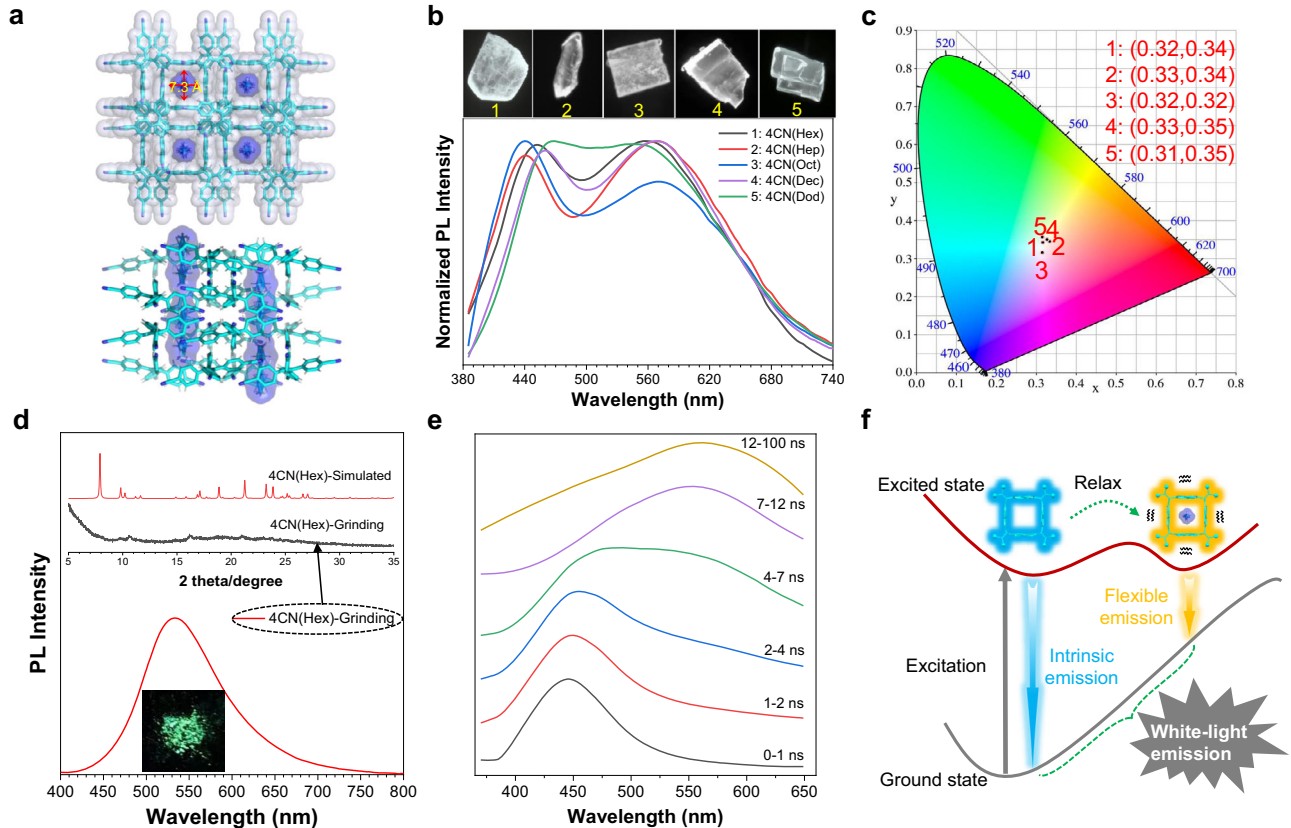

**Fig. 4 Multiple white-light emissive crystals of 4CN. a** Packing diagram of 4CN(Hex) along the [001] direction and [100] direction. The dimensions of the cavity are highlighted by red arrows. **b** Normalized fluorescent emission spectra of 4CN(Hex), 4CN(Hep), 4CN(Oct), 4CN(Dec) and 4CN(Dod) crystals. Inset: fluorescent images of 1 for 4CN(Hex), 2 for 4CN(Hep), 3 for 4CN(Oct), 4 for 4CN(Dec) and 5 for 4CN(Dod) crystals. **c** The corresponding CIE chromaticity coordinates in CIE-1931 chromaticity diagram for 1–5. **d** PL spectra of 4CN(Hex) after grinding. Inset: the PXRD patterns of simulated 4CN(Hex) and its ground sample. **e** Time-resolved emission spectra of 4CN(Hex) crystal. $\lambda_{ex} = 343$ nm. **f** Proposed mechanism for white light emitting crystals.

PXRD profile of these five crystals with that of 4CN(ET1), they were also classified as **G2** group (Supplementary Fig. 13). The white-light emission bands centered at 441–469 nm and 549–571 nm with a relatively high quantum yield of 43% for 4CN(Hex) (Fig. 4b and Supplementary Table 1). As shown in Fig. 4c, the purity of white-light emission is estimated by the chromaticity diagram, in which the CIE coordinates are found to be (0.32, 0.34), (0.33, 0.34), (0.32, 0.32), (0.33, 0.35), and (0.31, 0.35), respectively. Though the solvents in the HOFs are highly disordered, we successfully refine the structure of 4CN(Hex) and 4CN(Dod), which reveals that the alkanes are located in the center of the pore with C–H···π interactions involved (Fig. 4a; Supplementary Figs. 9 and 11e–j). The entrapped highly disordered solvents were further confirmed by $^1$H NMR spectra (Supplementary Figs. 16–20).

We further wonder to explore the mechanism of white-light emission. Considering that all the other non-white-light emitting crystals (**G1**, **G3**, and **G4**) exhibit blue (450 nm) to blue–green (500 nm) emission, the shorter-wavelength emission band (450–470 nm) of the white light emitting **G2** can be reasonably classified as the intrinsic emission (Supplementary Fig. 21). The remaining question is the origin of the yellow emission (550–570 nm). After heavily grinding, we found that the dual emission bands disappear accompanied with the emerging of a new peak at 537 nm (Fig. 4d). The PXRD spectra indicated the ground sample was almost an amorphous state (Fig. 4d). The molecules in amorphous state are in a relatively disordered and

flexible environment. The origin of the amorphous yellow emission was thus thought to be induced by a flexible environment. Inspired by this, we propose that the molecule in the framework is also located in a similar flexible environment that is derived from two factors: the soft molecular conformation and dynamic intermolecular interactions constitute the flexible skeleton; the flexibility derived from highly disorder long-chain alkanes in the crystals further renders the benzonitrile peripheries in the channel more space for movement in the excited state. This flexible environment can thus allow relaxation in the excited state to generate a longer-wavelength emission.

To further verify our speculation, the photoluminescence decay curves of all the white-light emissive crystals were firstly measured. Their lifetime profiles were very similar, whose dual emissions were all evaluated to be nanosecond scale. This indicates both the blue and yellow emissions can be ascribed to fluorescence (Supplementary Fig. 23). Noteworthy, the lifetimes of yellow emission peaks are about three or four times longer than those of blue emission peaks. For example, for 4CN(Hex) crystal, its lifetimes are 1.2 ns for 452 nm and 5.1 ns for 560 nm, respectively (Supplementary Fig. 23a). This confirms that the longer-wavelength emission may originate from the relaxation of the short-wavelength emission in the excited state. More direct evidence was provided by the time-resolved emission spectra (TRES) of the white light emissive crystals. Taken 4CN(Hex) crystal as an example, it is obvious that the earlier time gate spectrum (0–4 ns) is characterized by blue emission with no

contribution from the yellow region, which confirms the intrinsic character of blue emission (Fig. 4e). With the time gate moves to prolonged time (4–7 ns), the PL spectrum exhibits a broad emission with two peaks located at 460 and 550 nm. However, the contribution from yellow emission becomes dominative during 7–100 ns time gate. The progressively weakened shorter-wavelength emission and intensified longer-wavelength emission confirms that the yellow emission should be relaxed from the blue emission. We can thus conclude, for the white-light with dual emission, its blue emission peak comes from the intrinsic emission of the framework, whereas the yellow one should be derived from the flexible solvent and dynamic framework-induced excited state relaxation (Fig. 4f).

**Framework switching.** Considering the frameworks are flexible, which may thus potentially respond to different external stimuli to realize multimode structural transformations. The thermal stability of 4CN(Hex) was firstly studied. Its TGA profile showed a fast weight loss of about 8% from ca. 110 °C to ca. 123 °C, which indicated escaping of solvents at this temperature range (Fig. 5a). This platform was further maintained until the decomposition of the material was beyond 335 °C. In the differential scanning calorimetry (DSC) profile from 30 to 300 °C, the porous 4CN(Hex) displayed an obvious endothermic peak at about 113 °C in the first heating process which is attributed to the *n*-hexane solvents releasing (Fig. 5b). Notably, no other exothermic or endothermic peak appeared during heating. Though it seems that no phase transition occurred upon thermal stimulation, desolvation process may break the interactions between solvent molecules and framework, which can thus induce structure transformation to a denser phase. To confirm this hypothesis, variable-temperature PXRD for 4CN(Hex) were measured from

20 to 150 °C (Fig. 5c). Its PXRD profile showed no obvious change below 110 °C. From 110 to 114 °C, the diffraction peak of the PXRD profile changed abruptly between 10° and 25°, accompanied by some new peaks appearing and meanwhile, some initial peaks, such as the peak at $2\theta = 7.82°$ disappearing. This phenomenon should be ascribed to a crystalline phase transition. Upon further heating, a new PXRD pattern formed which was exactly the same as that of the aforementioned 4CN(Non1) in Fig. 2b(i). This means, in response to guest removal, solvated and kinetic stable 4CN(Hex) HOF could convert to a denser and thermal stable 4CN(Non1) conformation. The gas adsorption-desorption experiments further confirmed the flexibility of the framework. After degassing at 110 °C under a high vacuum for 12 h, the sample showed low uptake capacity for $N_2$ with BET surface area of 3.0 $m^2g^{-1}$ at 77 K and 1 bar (Supplementary Fig. 24a). PXRD profile of 4CN(Hex) after gas adsorption-desorption was also consistent with that of simulated nonporous 4CN(Non1) (Supplementary Fig. 24b). This confirms that the kinetically stable porous crystals can transform into thermo-dynamically stable nonporous crystalline state.

Crystal analysis provide more detailed information about the transformation process. The square-shaped pore in 4CN(Hex) is assembled by four 4CN molecules, through eight pairs of intermolecular C–H···N hydrogen bonds between the aromatic C–H groups and nitrogen atoms from the cyano-groups with distances in the range of 2.675–2.754 Å (Supplementary Fig. 11e). The *n*-hexane guest is weakly entrapped in the channel through C–H···π interactions. Thus, heating can induce the losing of *n*-hexane in accompanied with the rearrangement of the packing to the nonporous 4CN(Non1) conformation. The strong π–π stacking interactions in 4CN(Non1) that can stabilize the packing should be one important driving force for this transformation. Similarly, thermal can also induce the release of ethanol from

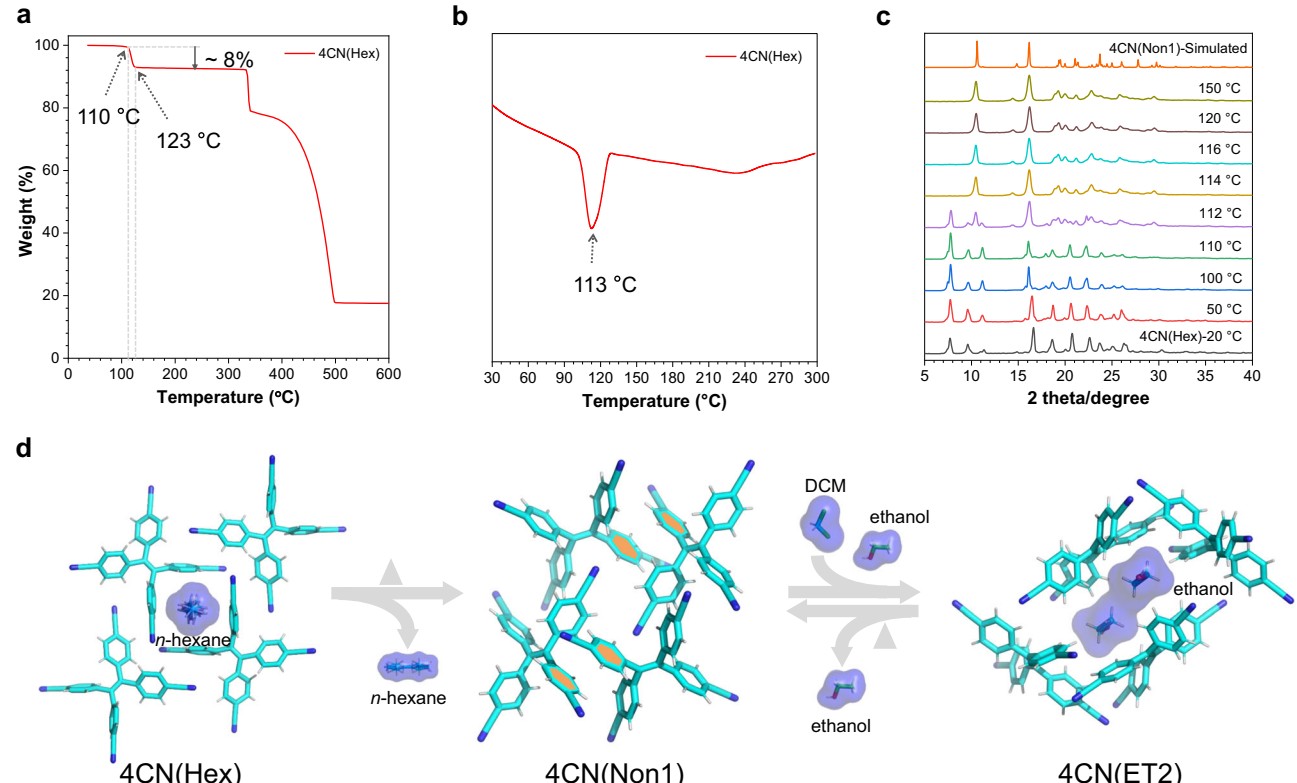

**Fig. 5 The transformation of 4CN(Hex). a** TGA, **b** DSC, and **c** variable-temperature PXRD of 4CN(Hex) crystals upon heating. **d** Structure transformation between porous 4CN(Hex) or 4CN(ET2) and nonporous 4CN(Non1) upon heating and solvent fuming. 'Δ' indicates heating.

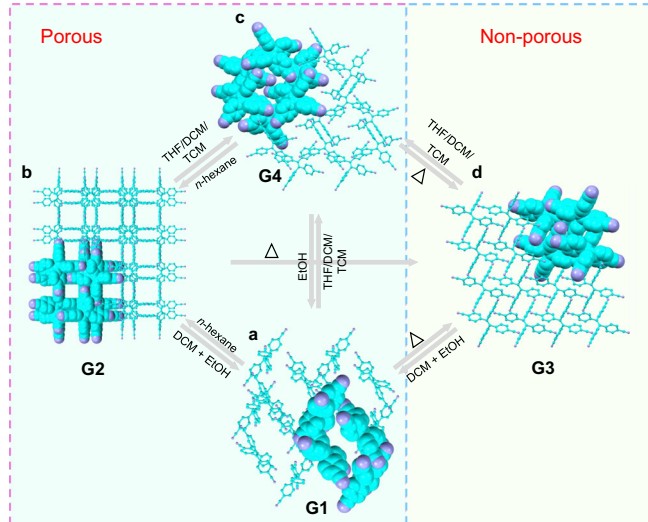

**Fig. 6 Multimode reversible transformations of 4CN. a** Packing diagram of **G1**: 4CN(ET2) and 4CN(MT). **b** Packing diagram of **G2**: 4CN(ET1), 4CN(Hex), 4CN(Hep), 4CN(Oct), 4CN(Dec), and 4CN(Dod). **c** Packing diagram of **G4**: 4CN(THF), 4CN(DCM), and 4CN(TCM). **d** Packing diagram of **G3**: 4CN(Non1) and 4CN(Non2). '∆' indicates heating.

4CN(ET2) to deliver 4CN(Non1). Moreover, we found almost all the other HOFs could finally transform to the 4CN(Non1) conformation upon heating as demonstrated by TGA, DSC, and PXRD (Supplementary Figs. 25–27). It is worth mentioning that if the freshly prepared crystals of 4CN(ET1) and 4CN(ET2) are placed in the mother liquor for another 2 weeks, they can also finally transform into 4CN(Non1) (Supplementary Fig. 28). This again confirms that 4CN(Non1) is the most stable one among the multiple 4CN crystalline phases. What's more, the nonporous 4CN(Non1) can be converted back to the 4CN(ET2) framework by fuming with dichloromethane/ethanol vapor (Fig. 5d and Supplementary Fig. 29). In combination with the aforementioned PXRD profiles in Fig. 2b and Supplementary Fig. 12, we know that the conformation in **G3** can transform into that in **G1** and **G4** (Fig. 6). This means reversible crystalline phase transformation can be realized between solvent-free nonporous structure (**G3**) and the solvent-containing porous structure (**G1** and **G4**) (Supplementary Figs. 29–32). It should be noted, though 4CN(Non1) is not crystallized in a channel-forming manner, in current opinion, the presence of interstitial or latent voids can also be prerequisite of the material to be considered porous. Thus, we believe concerted dynamic processes occur within the host structure facilitates breathing-like diffusion of different solvents via transient channels to generate various porous sturcture[70–72].

Moreover, we found that the 11 different porous structures (**G1**, **G2**, and **G4**) could switch between each other upon different vapor-triggering (Fig. 6). For example, after fuming 4CN(THF) crystals of **G4** with ethanol for 1 day, they can be fully transformed to 4CN(ET2) of **G1**, as proved by the PXRD pattern (Supplementary Fig. 33). If further fuming the resulted 4CN(ET2) sample with tetrahydrofuran, its conformation can completely turn back to the original 4CN(THF) (Supplementary Fig. 33), which confirms the switchable transformation between **G1** and **G4**. Similarly, the switching between **G2** and **G4** can also be realized as represented by the reversible PXRD pattern of the 4CN(THF) and 4CN(Hex) with corresponding solvents fuming (Supplementary Fig. 34). Besides, similar transformations can occur in other porous crystals that contain solvents (Supplementary Figs. 35–38). The multimode reversible transformations can not only be proved by PXRD, but also monitored by fluorescence (Supplementary Figs. 39–44).

The fluorescence showed no signs of fatigue under modest cycling demonstrated that these transitions were reversible (Supplementary Figs. 39b, 41b, 42b, and 43b). Based on the above observations, we can conclude that not only the reversible transformation between porous and nonporous structures is realized, but also the switching between different porous structures can be completed. The synergy of soft intermolecular interactions and flexible conformations give HOFs enough dynamic to be reversibly transformed through the direct exchange of solvent molecules.

## Discussion

In summary, we have successfully constructed multiple yet switchable HOFs based on a simple $C_2$ symmetry tecton 4CN. Specifically, eleven kinetic-stable porous forms with various shapes and two thermo-stable nonporous structures with rare perpendicular conformation are obtained. Based on the synergy of dynamic intermolecular interactions and flexible molecular conformation, multimode reversible structural transformations between porous and nonporous or between different porous forms can be realized under the stimulation of vapor and temperature. The AIE character of 4CN endows the assemblies with variable emission ranges from blue to green and thus visible switching process. Furthermore, the introduction of soft guests, such as five linear alkanes of different lengths, to the channel of flexible HOFs, can facilitate the excited state relaxation from intrinsic blue emission to yellow emission, which combined to produce pure white-light emission. This flexible environment-induced excited state relaxation represents an alternative strategy for white-light emission generation. The synergy of dynamic intermolecular interactions and flexible molecular conformation should also pave an avenue toward multi-switchable smart materials.

## Methods

**Preparation of the single crystals of 4CN.** The single crystals of 4CN(MT), 4CN(ET2), 4CN(THF), 4CN(DCM), and 4CN(TCM) were grown by slow solvent evaporation of a saturated solution of 4CN in methanol or ethanol or tetrahydrofuran or dichloromethane or trichloromethane at room temperature for 1 week, respectively. The single crystals of 4CN(Non1) and 4CN(Non2) were grown at room temperature for 1 week by slow vapor diffusion of cyclohexane into their tetrahydrofuran solution. The single crystals of 4CN(Hex), 4CN(Hep), 4CN(Oct), 4CN(Dec), and 4CN(Dod) were grown at room temperature for 1 week by slow vapor diffusion of *n*-hexane or *n*-heptane or *n*-octane or *n*-decane or *n*-dodecane into their acetone (toluene) solution, respectively. The single crystals of 4CN(ET1) were grown by slow solvent evaporation of a saturated solution of 4CN in dichloromethane/ethanol (2:1) mixture at room temperature[68].

## Data availability

The authors declare that all data supporting the findings of this study are available within this article and Supplementary Information files, and also are available from the authors upon request. The X-ray crystallographic coordinates for structures reported in this study have been deposited at the Cambridge Crystallographic Data Centre (CCDC), under deposition numbers CCDC 2102622, 2111294, and 2090902–2090911. These data can be obtained free of charge from The Cambridge Crystallographic Data Centre via www.ccdc.cam.ac.uk/getstructures. Source data are provided with this paper.

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

## Acknowledgements

P.W. thanks the support from the National Natural Science Foundation of China Grants (22001006), the Open Fund of Guangdong Provincial Key Laboratory of Luminescence from Molecular Aggregates and South China University of Technology (2019B030301003), the Innovation and Entrepreneurship Project of Overseas Returnees in Anhui Province (2020LCX017) and The Open Project of Key Laboratory of Structure and Functional Regulation of Hybrid Materials of Anhui University, Ministry of Education. We thank Dr. Jianzhuang Chen from East China University of Science and Technology for helping to do variable-temperature PXRD. We thank Prof. Guoqing Zhang, Prof. Xuepeng Zhang, and Dr. Xiancheng Nie from University of Science and Technology of China for helping to measure lifetime and quantum yields. We thank Prof. Yu Fang and Prof. Taihong Liu from Shaanxi Normal University for helping to do TRES.

## Author contributions

Y.S., P.W., and B.Z.T. conceived and designed the experiments. Y.S., Y.D., G.X., C.C. X.S., Z.Z., and P.W. performed the experiments. S.W. synthesized the 4CN compound. W.T. collected and refined all the crystal data. J.G. performed the DFT calculation. S.X., Z.H., and other authors were all involved in the analyses and interpretation of data. Y.S. wrote the paper with the help of P.W. All authors discussed the results and commented on the manuscript.

## Competing interests

The authors declare no competing interests.
