## [Peer Review File · Nature Communications]

Multiple yet switchable hydrogen-bonded organic frameworks
with white-light emissionREVIEWER COMMENTS

Reviewer #1 (Remarks to the Author):

Comments on: Multiple yet switchable porous molecular crystals with white-light emission
In the present study, authors have developed eleven kinetically stable porous and two thermodynamically stable nonporous fluorescent molecular crystals of simple molecular tectons, i.e., tetracyano substituted tetraphenylethylene. Just by grinding and fuming the molecular crystals with different organic solvents, they have demonstrated the tuning of dynamic intermolecular interactions and flexible molecular conformation that leads to the change in the fluorescent properties (blue to green emission). On the other hand, the introduction of flexible alkanes with varying lengths, e.g, hexane, heptane, etc., into the porous crystals can lead to white light emission. The study is unique, and the structural characterizations of switchable porous molecular crystals have been carried with great detail. Hence, publication of this manuscript is recommended. However, authors are requested to address the below points in the revised manuscript to further augment the clarity and quality, particularly focusing on the physical picture behind white light emission.

- (i) Specific BET surface areas of any of the kinetically stable porous crystals and thermodynamically stable nonporous crystals can be checked.
- (ii) The following paper can be cited against the following statement. "Virtual porosity" was observed in the crystal structure, with a total solvent accessible void at 17.2% of its volume estimated by PLATON analysis using a 1.2 Å probe." (Crystal porosity and the burden of proof, *Chem. Commun.*, 2006, 1163–1168).
- (iii) What is the typical pore size of the porous molecular crystals. Please, modify Figure 3a by designating the host matrix and the guest (hexane) in a space-filled model.
- (iv) The physical picture behind the white light emission is not fully clear to me. The concept of the 'super-flexible excited state' or the 'flexible environment-induced super-flexible excited state relaxation' requires further analysis and elaboration in the main text. Some of the time-resolved emission data and analysis should be brought to the main text for better clarity of the general readers.
- (v) Further, systematic fluorescence decay analysis and time-resolved emission spectroscopy (TRES) measurements will be helpful to bring the physical picture behind the white light emission.
- (vi) In the introduction part, special emphasis is to be given for single-component white light emission in molecular materials. The following article can be cited, *Chem. Eur. J.*, 2020, 26, 5557-5582.

Reviewer #2 (Remarks to the Author):

This paper describes the creation of porous molecular crystals (PMCs), so-called inclusion crystals, using 4CN compounds that exhibit AIEE properties and the creation of solid-state light-emitting materials depending on the guest molecules. In particular, it is interesting to note that the alkane-incorporated crystals exhibit white luminescence. It is commendable that the authors have achieved two-color luminescence by combining intrinsic luminescence and flexible luminescence by guest inclusion, and are close to elucidating the luminescence mechanism.

The paper itself is written in a very attractive manner, but other than the fact that white luminescence was obtained, it seems to describe the behavior of well-known luminescent inclusion crystals. It contains an introduction on PMCs, but hardly mentions any references on white light emissive materials, which is the main topic of the paper. The 4CN used in this

study is also a previously reported, and the key single crystal structure (4CN(ET1)) with an open channel structure is also previously reported (Ref. 59). Therefore, the novelty of the research is not remarkably high. Even though the paper describes the optical properties of the crystalline material, the crystals shown in Figure 1a has only simple fluorescence spectra and photograph data, which cannot be compared with 4CN(Hex), which shows white light emission. The crystal phase transformation of 4CN in the latter half of this paper are also only evaluated structurally by PXRD, and there is a lack of data on how much the actual optical properties have changed. Therefore, this reviewer judge that this manuscript is not up to the level of Nature Communication, which requires a very high level, and recommend that the paper to another paper.

The following are some comments to improve this paper.

- (1) First of all, please clarify the difference between porous molecular crystals (PMCs) and conventional inclusion crystals.
- (2) There is a lack of reference literature on white light-emitting materials using organic compounds in the introduction. Please add.
- (3) The background of the absorption spectrum of the crystal shown in Fig. 1(a) is very large; is this due to the transmission method or diffuse reflectance spectroscopy? Please describe it in detail.
- (4) Optical properties such as quantum yield and luminescence lifetime for the crystalline materials are not listed. At least for 4CN(ET2), 4CN(MT), 4CN(ET1), 4CN(Non2), 4CN(DCM), 4CN(TCM), and 4CN(THF) shown in Fig. 1(a), please describe the results in the Supplementally Information. Additionally, the data for 4CN(Hep), 4CN(Oct), 4CN(Dec), and 4CN(Dod), should be included in the Supplementally Information.
- (5) Since it is difficult to understand the composition ratio of 4CN to guest in each crystal, please summarize the data in the Supplementally Information.
- (6) For the multimode reversible transformations of 4CN shown in Fig. 5, please check not only the PXRD measurement but also the optical measurement data.
- (7) The description of the computational chemistry is inadequate and the results cannot be reproduced. The only information provided is that the Gaussian 09 program was used. For example, in Supplementary Figure S4(b), it is unclear how the crystal structure was stabilized, so please clarify this point.
- (8) In Supplementary Figure S5, the vertical axis is compared in terms of total energy (eV), but it is difficult to understand. In general, it is often compared in kcal/mol or kJ/mol, so please correct it to that.

Reviewer #3 (Remarks to the Author):

The manuscript reports that the molecule of tetracyano-functionalized TPE derivative could crystallized in multiple crystal structures with varied fluorescent emission, and the molecular crystals can reversibly transfer by organic vapor fuming. Furthermore, after inclusion of organic vapor, such as hexane, the crystals are dual fluorescence emission and present

white-light emitting PMCs. Although the observation is quite interesting, the quantitative analysis of the amount of organic vapor to the light emission properties are not well-established. Further investigation is requested before I can recommend for acceptance.

1. Why the crystals are obvious dual emitted after hexane or heptane inclusion, and how the guests affect the relaxation of host. The difference of relax behavior between crystals inclusion of hexane and THF or other guests.

2. Maybe the author should further illustrate why the stronger π - π interaction result in the emission redshift.

3. Emission enhancement and red-shifting of sample by adding n-hexane to acetone may not illustrate the flexibility of molecule. Various molecular conformation is fixed as the rapid aggregation process and form nonporous solid.

4. The author claims "porous molecular crystals", but the porosity was not established by gas or vapor adsorption isotherm. 5. It is not clear that how stable and reversible is the skeleton. How many cycles can the crystals perform reversible light emission?

Point-by-point response to the reviewers' comments:

General response to all the three reviewers: In view of the comments of the three reviewers, reviewer 1 (comment 1) and reviewer 3 (comment 4) raised questions about the BET of the porous crystals, reviewer 2 (comment 1) asked us to clarify the difference between porous molecular crystals (PMCs) and inclusion crystals. All these comments can be classified as the question related to porosity. As expected, BET result indicates the pores of our materials is not robust. However, in this work we focus on the “flexible and dynamic” nature of the crystals. This character not only brings about multiple and switchable structural transformation between the 13 crystals, but also facilitates the flexible environment-induced relaxation in the excited state which finally leads to white-light emission. PMCs focuses much more on robustness, which may not be the most suitable name for our materials. Thus, we decided to replace the term “porous molecular crystals” with “hydrogen-bonded organic frameworks” (HOFs) in consideration of the following three reasons: 1) The main intermolecular interactions in the crystals is hydrogen bonds and HOF is a more specific concept compared with that of inclusion crystals; 2) the definition of HOFs does not pose requirement on the robustness of the pore when the solvent is removed, which fits more closely with the character of our materials; 3) The channels of the crystals play an important role in this work, especially for the generation of white-light emission. So we would like to change the title of this manuscript from “Multiple yet switchable porous molecular crystals with white-light emission” to “Multiple yet switchable hydrogen-bonded organic frameworks with white-light emission”.

Reviewer 1

The study is unique, and the structural characterizations of switchable porous molecular crystals have been carried with great detail. Hence, publication of this manuscript is recommended.

Response: We would like to thank you for your positive comments and valuable suggestions to improve our manuscript.

Comment 1: Specific BET surface areas of any of the kinetically stable porous crystals and thermodynamically stable nonporous crystals can be checked.

Response: Thanks for the valuable comments. As we have mentioned in “General response to all the three reviewers”, the porous structure in this manuscript is not robust. For example, for kinetically stable crystals 4CN(Hex) after degassing at 110 °C under high vacuum for 12 h, it shows low uptake capacity for N₂ with BET surface area of 3.0 m²g⁻¹ at 77 K and 1 bar. PXRD profile of 4CN(Hex) after gas adsorption-desorption is consistent with that of simulated nonporous 4CN(Non1). This confirms that the kinetically stable porous crystals can transform into thermodynamically stable nonporous crystalline state during degassing (Figure S24). The relevant description was added in the main text of the revised manuscript: “**The gas adsorption-desorption experiments further confirmed the flexibility of the framework. After degassing at 110 °C under high vacuum for 12 h, the sample showed low uptake capacity for N₂ with BET surface area of 3.0 m²g⁻¹ at 77 K and 1 bar (Supplementary Fig. S24a). PXRD profile of 4CN(Hex) after gas adsorption-desorption was also consistent with that of simulated nonporous 4CN(Non1) (Figure S24b). This confirms that the kinetically stable porous crystals can transform into thermodynamically stable nonporous crystalline state.**”

Supplementary Figure S24. (a) Sorption isotherms measured on 4CN(Hex) for N₂ at 77 K. (b) The PXRD spectra of simulated 4CN(Hex), 4CN(Non1) and 4CN(Hex) after gas adsorption-desorption.

Comment 2: The following paper can be cited against the following statement. “Virtual porosity” was observed in the crystal structure, with a total solvent accessible void at 17.2% of its volume estimated by PLATON analysis using a 1.2 Å probe. (Crystal porosity and the burden of proof, Chem. Commun., 2006, 1163-1168).

Response: Thank you for pointing out this mistake. We have corrected the statement and this reference is cited. The relevant description was added in the main text of the revised manuscript: “The crystal structure with a total solvent accessible void at 17.2% of its volume was estimated by PLATON analysis using a 1.2 Å probe^{38,69}”.

Comment 3: What is the typical pore size of the porous molecular crystals. Please, modify Figure 3a by designating the host matrix and the guest (hexane) in a space-filled model.

Response: Thanks for the valuable comments. Figure 3a has been modified by designating the host matrix and the guest (hexane) in a space-filled model and the revised figure has been added in the main text.

Fig. 3a Multiple white-light emissive crystals of 4CN. (a) Packing diagram of 4CN(Hex) along the [001] direction and [100] direction. The dimensions of the cavity are highlighted by red arrows.

Comment 4: The physical picture behind the white-light emission is not fully clear to me. The concept of the “super-flexible excited state” or the “flexible environment-induced super-flexible excited state relaxation” requires further analysis and elaboration in the main text. Some of the time-resolved emission data and analysis should be brought to the main text for better clarity of the general readers.

Comment 5: Further, systematic fluorescence decay analysis and time-resolved emission spectroscopy (TRES) measurements will be helpful to bring the physical picture behind the white light emission.

Response: Thanks for the valuable comments. Both comment 4 and comment 5 are about the mechanism of white-light emission. We are deeply sorry for the confusion caused by the concept to reviewers. For the white-light with dual emission, its blue emission peak comes from the intrinsic emission of the framework whereas the yellow one can be ascribed to the flexible environment-induced excited state relaxation. The flexible environment is mainly originated from two factors: the soft molecular conformation and dynamics intermolecular interactions constitute the flexible skeleton; the flexibility derived from highly disorder long-chain alkanes in the crystals

further renders the benzonitrile peripheries in the channel more space for movement in the excited state. To avoid confusing, the phrase of “super-flexible excited state” has been deleted and the phrase of “flexible environment-induced super-flexible excited state relaxation” has been changed to “flexible environment-induced excited state relaxation”. The systematic fluorescence decay analysis of all the white-light emissive crystals has been measured (Supplementary Fig. S23). The yellow emission exhibits longer lifetime than that of blue one, which indicates its possibility of relaxing from short-wavelength emission in the excited state. Taking 4CN(Hex) as an example, we have provided its time-resolved emission spectra in Fig. 3e. Upon increasing the gating time, the progressively weakened shorter-wavelength emission and intensified longer-wavelength emission confirms directly that the yellow emission should be relaxed from the blue emission. Moreover, for better understanding the physical picture behind white-light emission, the whole part of this section has been rewritten:

“We further wonder to explore the mechanism of white-light emission. Considering that all the other non-white-light emitting crystals (G1, G3 and G4) exhibit blue (450 nm) to blue-green (500 nm) emission, the shorter-wavelength emission band (450-470 nm) of the white-light emitting G2 can be reasonably classified as the intrinsic emission (Supplementary Fig. S21). The remaining question is the origin of the yellow emission (550-570 nm). After heavily grinding, we found that the dual emission bands disappear accompany with the emerging of a new peak at 537 nm (Fig. 3d). The PXRD spectra indicated the ground sample was almost an amorphous state (Fig. 3d). The molecules in amorphous state are in a relatively disordered and flexible environment. The origin of the amorphous yellow emission was thus thought to be induced by a flexible environment. Inspired by this, we propose that the molecule in the framework is also located in a similar flexible environment that is derived from two factors: the soft molecular conformation and dynamic intermolecular interactions constitute the flexible skeleton; the flexibility derived from highly disorder long-chain alkanes in the crystals further renders the benzonitrile peripheries in the channel more space for movement in the excited state. This flexible environment can thus allow relaxation in the excited state to generate a longer-wavelength emission.

To further verify our speculation, the photoluminescence decay curves of all the white-light emissive crystals were firstly measured. Their lifetime profiles were very similar whose dual emission were all evaluated to be nanosecond scale. This indicates both the blue and yellow emission can be ascribed to fluorescence (Supplementary Fig. S23). Noteworthy, the lifetimes of

yellow emission peaks are about three or four times longer than those of blue emission peaks. For example, for 4CN(Hex) crystal, its lifetimes are 1.2 ns for 452 nm and 5.1 ns for 560 nm, respectively (Supplementary Fig. S23a). This confirms that the longer-wavelength emission may originate from the relaxation of the short-wavelength emission in the excited state. More direct evidence was provided by time-resolved emission spectra (TRES) of the white-light emissive crystals. Taken 4CN(Hex) crystal as an example, it is obvious that the earlier time gate spectrum (0-4 ns) is characterized by blue emission with no contribution from yellow region, confirms the intrinsic character of blue emission (Figure 3e). With the time gate moves to prolonged time (4-7 ns), the PL spectrum exhibits a broad emission with two peaks located at 460 and 550 nm. However, the contribution from yellow emission becomes dominative during 7-100 ns time gate. The progressively weakened shorter-wavelength emission and intensified longer-wavelength emission confirms that the yellow emission should be relaxed from the blue emission. We can thus conclude, for the white-light with dual emission, its blue emission peak comes from the intrinsic emission of the framework whereas the yellow one should derived from the flexible solvent and dynamic framework-induced excited state relaxation (Fig. 3f).”

Fig. 3 (e) Time-resolved emission spectra of 4CN(Hex) crystal. $\lambda_{\text{ex}} = 343$ nm.

Supplementary Figure S23. Fluorescence decay curves of white-emitting (a) 4CN(Hex), (b) 4CN(Hep), (c) 4CN(Oct), (d) 4CN(Dec) and (e) 4CN(Dod) crystals.

Comment 6: *In the introduction part, special emphasis is to be given for single-component white light emission in molecular materials. The following article can be cited, Chem. Eur. J., 2020, 26, 5557-5582.*

Response: Thanks for the valuable comments. We have added the suggested reference (Ref 61) and some other related references in Refs 58-60 and 62-66. Meanwhile, considering white-light emission is one of the main topics of this work, we strongly agree with all the reviewers' comments and has added a new paragraph related to organic white light emission materials in the introduction part of the revised manuscript:

“As we all know, organic white-light-emitting materials have attracted a lot of research attention due to their important application prospects. So far, there are many reports available on the construction of white-light-emitting systems, including nanostructures⁵⁸, supramolecules⁵⁹, small molecules,^{60, 61} hydrogels⁶² and so on. However, developing novel and simple strategies for preparing white-light-emitting materials is still highly desirable⁶³⁻⁶⁶. Frameworks with channels

can accommodate guests of matching size through non-covalent bond interactions. The characteristics of the guests will have a non-negligible effect on the luminescence of the frameworks^{49, 55}. Thus, the combination of fluorescence and switchable HOFs induced by guests may provide a potential way for the construction of tunable luminescent materials, including white-light-emitting materials.”

58. Wang, Z., et al. All-Copper Nanocluster Based Down-Conversion White Light-Emitting Devices. *Adv. Sci.* **3**, 1600182 (2016).
59. Li, D., Wang, J. & Ma, X. White-Light-Emitting Materials Constructed from Supramolecular Approaches. *Adv. Optical Mater.* **6**, 1800273 (2018).
60. Chen, Z., Ho, C. L., Wang, L. & Wong, W. Y. Single-Molecular White-Light Emitters and Their Potential WOLED Applications. *Adv. Mater.* **32**, 1903269 (2020).
61. Kundu, S., Sk, B., Pallavi, P., Giri, A. & Patra, A. Molecular Engineering Approaches Towards All-Organic White Light Emitting Materials. *Chem. Eur. J.* **26**, 5557-5582 (2020).
62. Wang, J., Tang, F., Wang, Y., Liu, S. & Li, L. Tunable Single-Molecule White-Light Emission in Stimuli-Responsive Hydrogel. *Adv. Optical Mater.* **8**, (2020).
63. He, Z., et al. White light emission from a single organic molecule with dual phosphorescence at room temperature. *Nat. Commun.* **8**, 416 (2017).
64. Yang, Q. Y. & Lehn, J. M. Bright white-light emission from a single organic compound in the solid state. *Angew. Chem. Int. Ed.* **53**, 4572-4577 (2014).
65. Zhou, C., et al. Ternary Emission of Fluorescence and Dual Phosphorescence at Room Temperature: A Single-Molecule White Light Emitter Based on Pure Organic Aza-Aromatic Material. *Adv. Funct. Mater.* **28**, 1802407 (2018).
66. Xie, Z., et al. White-light emission strategy of a single organic compound with aggregation-induced emission and delayed fluorescence properties. *Angew. Chem. Int. Ed.* **54**, 7181-7184 (2015).

Reviewer 2

In particular, it is interesting to note that the alkane-incorporated crystals exhibit white luminescence. It is commendable that the authors have achieved two-color luminescence by combining intrinsic luminescence and flexible luminescence by guest inclusion, and are close to elucidating the luminescence mechanism.

Response: We would like to thank you for your positive comments on the white-light emission and valuable suggestions to improve our manuscript. After providing more direct evidences, such as the time-resolved emission spectra (TRES) based on the reviewers' suggestions, this mechanism behind white-light emission becomes more reliable and solid.

The paper itself is written in a very attractive manner, but other than the fact that white luminescence was obtained, it seems to describe the behavior of well-known luminescent inclusion crystals. The 4CN used in this study is also a previously reported, and the key single crystal structure (4CN(ET1)) with an open channel structure is also previously reported (Ref. 59). Therefore, the novelty of the research is not remarkably high.

Response: Thank you for your valuable comments. Broadly speaking, the crystals reported in this manuscript can be classified as luminescent inclusion crystals. However, in this work, we mainly focus on their multiple and controllable transformations based on their flexibilities. Moreover, a new strategy of combining intrinsic emission and flexible emission for organic white-light emission is another novelty of this work.

We are sorry for the 4CN(ET1) mentioned in our manuscript was first reported in the article (*Chem. Commun.* 2013, **49**, 3961), which affected your judgment on the novelty of our work to a certain extent. The *Chem. Commun.* article reports a covalent triazine-based organic framework (CTF) constructed through transformation of 4CN under Lewis acidic at 400 °C. In our manuscript, we have successfully constructed multiple yet switchable HOFs based on 4CN molecules. It should be noted that 4CN molecule reported in the *Chem. Commun* paper is an intermediate of the target product. Meanwhile, 4CN(ET1) is only one of the thirteen crystals reported in our work. Moreover, taking advantage of the flexibility of 4CN molecule and the open channel of 4CN(ET1) crystal, we realize the replacement of entrapped ethanol guest by soft long-chain alkanes and obtain five

white-light emitting crystals, named 4CN(Hex), 4CN(Hep), 4CN(Oct), 4CN(Dec) and 4CN(Dod). This is quite different from the green emissive 4CN(ET1) (The *Chem. Commun.* paper did not talk about fluorescence). The mechanism behind the white-light emission was well illustrated. This flexible environment-induced excited state relaxation represents a new and unique strategy for white-light emission generation. More specific highlights of this work are listed as below:

(1) This is one of the most flexible HOFs reported so far. Thirteen different kinds of kinetic-stable or thermal-stable conformations are constructed solely from an extremely simple molecule.

(2) The flexible molecular conformations and dynamic intermolecular interactions in combination with tunable fluorescence result in reversible and visible multi-transformations between the thirteen crystals.

(3) A new and general strategy for white-light emission is proposed: synergy of flexible framework and soft long-chain guests can trigger the relaxation of excited states to deliver dual emission.

It contains an introduction on PMCs, but hardly mentions any references on white light emissive materials, which is the main topic of the paper.

Response: Thank you for pointing out this oversight. We have added a new paragraph related to the construction of white light emissive materials in the introduction part:

“As we all know, organic white-light-emitting materials have attracted a lot of research attention due to their important application prospects. So far, there are many reports available on the construction of white-light-emitting systems, including nanostructures⁵⁸, supramolecules⁵⁹, small molecules,^{60, 61} hydrogels⁶² and so on. However, developing novel and simple strategies for preparing white-light-emitting materials is still highly desirable⁶³⁻⁶⁶. Frameworks with channels can accommodate guests of matching size through non-covalent bond interactions. The characteristics of the guests will have a non-negligible effect on the luminescence of the frameworks^{49, 55}. Thus, the combination of fluorescence and switchable HOFs induced by guests may provide a potential way for the construction of tunable luminescent materials, including white-light-emitting materials.”

Even though the paper describes the optical properties of the crystalline material, the crystals shown in Figure 1a has only simple fluorescence spectra and photograph data, which cannot be compared with 4CN(Hex), which shows white light emission.

Response: Thanks for the valuable comments. We have added Fig. S21 which contains the fluorescence spectra of eight non-white-light emitting crystals (solid lines) and five white-light emitting crystals (dots) in the supplementary information for better comparison.

Supplementary Figure S21. Normalized fluorescence spectra of the thirteen crystals.

Comment 1: First of all, please clarify the difference between porous molecular crystals (PMCs) and conventional inclusion crystals.

Response: Thanks for the valuable comments. As we have mentioned in “General response to all the three reviewers” at the very beginning, the porous structure in this manuscript is not robust. Our material is flexible and can not maintain the porosity after removing the solvent. After carefully checking, we found that porous molecular crystals (PMCs) focuses much more on robustness, which may be not the most suitable name for our materials. Inclusion crystals is a much broader concept than that of PMCs. So broadly speaking, the thirteen crystals reported in this manuscript can be classified as inclusion crystals. However, the channels of the crystals play an important role in this work, especially for the generation of white-light emission. Meanwhile, the

main intermolecular interactions in the crystals is hydrogen bonds and the definition of hydrogen-bonded organic frameworks does not pose requirement on the robustness of the pore when the solvent is removed, which fits more closely with the flexible character of our materials. So, to more accurately describe our materials, the term “porous molecular crystals” was replaced by “hydrogen-bonded organic frameworks” in our manuscript.

Comment 2: *There is a lack of reference literature on white light-emitting materials using organic compounds in the introduction. Please add.*

Response: Thanks for the valuable comments. We have added related references (Refs. 58-66) on organic white-light-emitting materials in the introduction part.

58. Wang, Z., et al. All-Copper Nanocluster Based Down-Conversion White Light-Emitting Devices. *Adv. Sci.* **3**, 1600182 (2016).
59. Li, D., Wang, J. & Ma, X. White-Light-Emitting Materials Constructed from Supramolecular Approaches. *Adv. Optical Mater.* **6**, 1800273 (2018).
60. Chen, Z., Ho, C. L., Wang, L. & Wong, W. Y. Single-Molecular White-Light Emitters and Their Potential WOLED Applications. *Adv. Mater.* **32**, 1903269 (2020).
61. Kundu, S., Sk, B., Pallavi, P., Giri, A. & Patra, A. Molecular Engineering Approaches Towards All-Organic White Light Emitting Materials. *Chem. Eur. J.* **26**, 5557-5582 (2020).
62. Wang, J., Tang, F., Wang, Y., Liu, S. & Li, L. Tunable Single-Molecule White-Light Emission in Stimuli-Responsive Hydrogel. *Adv. Optical Mater.* **8**, (2020).
63. He, Z., et al. White light emission from a single organic molecule with dual phosphorescence at room temperature. *Nat. Commun.* **8**, 416 (2017).
64. Yang, Q. Y. & Lehn, J. M. Bright white-light emission from a single organic compound in the solid state. *Angew. Chem. Int. Ed.* **53**, 4572-4577 (2014).
65. Zhou, C., et al. Ternary Emission of Fluorescence and Dual Phosphorescence at Room Temperature: A Single-Molecule White Light Emitter Based on Pure Organic Aza-Aromatic Material. *Adv. Funct. Mater.* **28**, 1802407 (2018).
66. Xie, Z., et al. White-light emission strategy of a single organic compound with aggregation-induced emission and delayed fluorescence properties. *Angew. Chem. Int. Ed.* **54**, 7181-7184 (2015).

***Comment 3:** The background of the absorption spectrum of the crystal shown in Fig. 1(a) is very large; is this due to the transmission method or diffuse reflectance spectroscopy? Please describe it in detail.*

Response: Thanks for the valuable comments. The large background of the absorption spectrum of the crystal is due to the diffuse reflectance spectroscopy. The caption of Figure 1(a) has been revised as “UV-Vis **diffuse reflectance** spectra (left) and PL spectra (right) of 4CN(Non1) crystals.” in the main text for better understanding.

***Comment 4:** Optical properties such as quantum yield and luminescence lifetime for the crystalline materials are not listed. At least for 4CN(ET2), 4CN(MT), 4CN(ET1), 4CN(Non2), 4CN(DCM), 4CN(TCM), and 4CN(THF) shown in Fig. 1(a), please describe the results in the Supplementally Information. Additionally, the data for 4CN(Hep), 4CN(Oct), 4CN(Dec), and 4CN(Dod), should be included in the Supplementally Information.*

Response: Thanks for the valuable comments. The quantum yields and fluorescence lifetime of all crystals have been measured and summarized in the **Supplementary Table S1** (highlight by yellow background). The fluorescence decay curves were also supplied (**Supplementary Figure S22-23**).

Supplementary Table S1. Summary of the dihedral angles (θ) between four phenyl groups central ethenyl group of different crystals, average dihedral angles (θ_{aver}), and their corresponding maximum emission wavelength (λ_{em}), quantum yield (Φ) and lifetimes (τ).

	θ_1	θ_2	θ_3	θ_4	θ_{aver}	λ_{em} (nm)	Φ	τ (ns)	
4CN(ET2)	65.58°	58.78°	56.88°	32.06°	53.32°	449	19.0%	2.1	G1
4CN(MT)	65.78°	60.11°	57.49°	36.56°	54.99°	449	12.5%	2.0	
4CN(ET1)	68.36°	68.36°	45.33°	45.33°	56.85°	470	37.5%	2.4	G2
4CN(Non2)	77.07°	76.86°	75.65°	64.10°	73.42°	471	16.6%	2.0	G3
4CN(Non1)	88.46°	88.46°	83.43°	83.43°	85.95°	482	44.8%	3.0	
4CN(DCM)	52.27°	43.70°	41.80°	38.63°	44.10°	481	21.6%	3.4	G4
4CN(TCM)	52.11°	45.74°	41.04°	36.43°	43.83°	497	65.4%	3.3	
4CN(THF)	53.57°	46.28°	42.91°	40.26°	45.76°	495	54.2%	3.3	
4CN(Hex)	65.92°	65.92°	44.12°	44.12°	55.02°	452,560	43.0%	1.2 (452 nm); 5.1 (560 nm)	G2 white light
4CN(Hep)	67.01°	67.01°	45.11°	45.11°	56.06°	441,569	31.6%	1.3 (441 nm); 5.1 (569 nm)	
4CN(Oct)	68.08°	68.08°	44.83°	44.83°	56.46°	441,571	28.6%	1.3 (441 nm); 4.5 (571 nm)	
4CN(Dec)	68.15°	68.15°	44.41°	44.41°	56.28°	460,569	27.6%	1.4 (460 nm); 4.0 (569 nm)	
4CN(Dod)	68.54°	68.54°	44.93°	44.93°	56.74°	469,549	27.2%	1.6 (469 nm); 4.6 (549 nm)	

Supplementary Figure S22. Fluorescence decay curves of non-white-emitting (a) 4CN(ET2), (b) 4CN(MT), (c) 4CN(ET1), (d) 4CN(Non1), (e) 4CN(Non2), (f) 4CN(DCM), (g) 4CN(TCM) and (h) 4CN(THF) crystals.

Supplementary Figure S23. Fluorescence decay curves of white-emitting (a) 4CN(Hex), (b) 4CN(Hep), (c) 4CN(Oct), (d) 4CN(Dec) and (e) 4CN(Dod) crystals.

Comment 5: Since it is difficult to understand the composition ratio of 4CN to guest in each crystal, please summarize the data in the Supplementally Information.

Response: Thanks for the valuable comments. For better comparison, the data including the composition ratios of 4CN to guests of thirteen crystals were summarized into two tables, **Supplementary Table S2** for non-white-light emitting crystals and **Supplementary Table S3** for white-light emitting crystals.

Supplementary Table S2. Crystal data for non-white-light emitting crystals.

Entry	4CN(ET2)	4CN(MT)	4CN(Non2)	4CN(Non1)	4CN(DCM)	4CN(TCM)	4CN(THF)
Empirical formula	(C₃₀H₁₆N₄) (C₂H₆O)	(C₃₀H₁₆N₄) (CH₄O)	C₃₀H₁₆N₄	C₃₀H₁₆N₄	(C₃₀H₁₆N₄) (CH₂Cl)	(C₃₀H₁₆N₄) (CHCl₃)	(C₃₀H₁₆N₄) (C₄H₄O)
Formula weight	478.53	460.48	432.47	432.47	517.39	551.83	500.54
Temperature/K	150	150	120	120	150	150	120
Crystal system	monoclinic	monoclinic	monoclinic	monoclinic	monoclinic	monoclinic	monoclinic
Space group	P2 ₁ /c	P2 ₁ /c	P2 ₁ /c	C2/c	P2 ₁ /c	P2 ₁ /c	P2 ₁ /c
a /Å	12.9854(6)	12.953	15.7974(8)	15.721	9.182	9.264	9.423
b /Å	15.0418(7)	14.863	9.7025(6)	9.835	12.862	12.832	12.902
c /Å	13.5120(6)	13.147	15.6259(8)	15.943	22.872	23.010	23.248
α /°	90	90	90	90	90	90	90
β /°	92.558(4)	90.15	98.297(4)	98.63	90.76	90.53	90.89
γ /°	90	90	90	90	90	90	90
Volume/Å ³	2636.6(2)	2531.0	2370.0(2)	2437.2	2701.2	2735.4	2825.9
Z	4	4	4	4	4	4	4
R ₁ /%	5.33	4.71	7.17	9.13	5.76	6.32	9.11
CCDC	2090909	2090907	2090905	2090904	2090910	2090903	2090906

Supplementary Table S3. Crystal data for white-light emitting crystals.

Entry	4CN(Hex)	4CN(Hep)	4CN(Oct)	4CN(Dec)	4CN(Dod)
Empirical formula ^a	(C ₃₀ H ₁₆ N ₄) (C ₆ H ₁₄) _x	(C ₃₀ H ₁₆ N ₄) (C ₇ H ₁₆) _x	(C ₃₀ H ₁₆ N ₄) (C ₈ H ₁₈) _x	(C ₃₀ H ₁₆ N ₄) (C ₁₀ H ₂₂) _x	(C ₃₀ H ₁₆ N ₄) (C ₁₂ H ₂₆) _x
Temperature/K	120	120	120	120	150
Crystal system	tetragonal	tetragonal	tetragonal	tetragonal	tetragonal
Space group	I4 ₁ /acd	I4 ₁ /acd	I4 ₁ /acd	I4 ₁ /acd	I4 ₁ /acd
a /Å	22.2889(9)	22.1994(10)	22.2301(7)	22.2896(13)	22.2596(13)
b /Å	22.2889(9)	22.1994(10)	22.2301(7)	22.2896(13)	22.2596(13)
c /Å	20.7869(9)	21.0846(10)	20.9280(5)	20.7847(11)	20.9592(18)
α /°	90	90	90	90	90
β /°	90	90	90	90	90
γ /°	90	90	90	90	90
Volume/Å ³	10326.8(9)	10390.8(11)	10342.1(7)	10326.4(13)	10385.1(15)
Z	16	16	16	16	16
R ₁ /%	6.56	7.83	9.89	6.81	8.33
CCDC	2090908	2090911	2090902	2111294	2102622

^ax: As the solvents in the channel are highly disordered, it is difficult to determine the exact ratio of solvent molecules to the framework.

Comment 6: *For the multimode reversible transformations of 4CN shown in Fig. 5, please check not only the PXRD measurement but also the optical measurement data.*

Response: Thanks for the valuable comments. The optical measurement data on the multimode reversible transformations of 4CN shown in Figure 5 has been completed (Supplementary Fig. S39-S44) and corresponds well with the PXRD results. The relevant figures were supplied in the Supplementary Figures S39-44 and the description was added in the main text of the revised manuscript: “The multimode reversible transformations can not only be proved by PXRD, but also monitored by fluorescence (Supplementary Fig. S39-44).”

Supplementary Figure S39. Normalized fluorescence spectra of reversible conversion between 4CN(ET2) (G1) and 4CN(Non1) (G3).

Supplementary Figure S40. Normalized fluorescence spectra of conversion from 4CN(Hex) (G2) to 4CN(Non1) (G3).

Supplementary Figure S41. Normalized fluorescence spectra of reversible conversion between 4CN(THF) (**G4**) and 4CN(Non1) (**G3**).

Supplementary Figure S42. Normalized fluorescence spectra of reversible conversion between 4CN(THF) (**G4**) and 4CN(Hex) (**G2**).

Supplementary Figure S43. (a) Normalized fluorescence spectra of reversible conversion between 4CN(THF) (**G4**) and 4CN(ET2) (**G1**).

Supplementary Figure S44. Normalized fluorescence spectra of (a) conversion from 4CN(ET1) (**G2**) to 4CN(ET2) (**G1**) and conversion from 4CN(ET2) (**G1**) to 4CN(Hex) (**G2**).

Comment 7: The description of the computational chemistry is inadequate and the results cannot be reproduced. The only information provided is that the Gaussian 09 program was used. For example, in Supplementary Figure S4(b), it is unclear how the crystal structure was stabilized, so please clarify this point.

Response: Thanks for the valuable comments. ONIOM model has been provided as **Supplementary Figure S4** for better understanding the calculation. The relevant description about calculation method was also added in the revised supplementary information: “**The ground state geometries were optimized using the density function theory (DFT) method with B3LYP hybrid functional at the basis set level of 6-31G(d,p) in gas phase. The combined quantum mechanics and molecular mechanics (QM/MM) method with two-layer ONIOM approach was used to simulate the properties in the solid state and the solid-phase computational model was built based on the X-ray crystal structure. The central molecule acted as the high layer (QM) at the B3LYP/6-31G(d,p) level, and the surrounding molecules were treated as the low layer (MM) using Universal Force Field (UFF). Beside, molecules of MM part are frozen during the QM/MM geometry optimizations for ground state. All these calculations were carried out in the Gaussian 16 package.**”

Supplementary Figure S4. ONIOM model: surrounding molecules are regarded as the low layer and the centered 4CN is treated as the high layer.

Comment 8: *In Supplementary Figure S5, the vertical axis is compared in terms of total energy (eV), but it is difficult to understand. In general, it is often compared in kcal/mol or kJ/mol, so please correct it to that.*

Response: Thanks for the valuable comments. The vertical axis in Figure S5 has been revised.

Supplementary Figure S5. Ground energy profile with different dihedral angles of 4CN in a gaseous environment by DFT calculation.

Reviewer 3

Although the observation is quite interesting, the quantitative analysis of the amount of organic vapor to the light emission properties are not well-established.

Response: Thanks for the valuable comments. The crystals were fumed with different time which were subsequent to ^1H NMR and fluorescent measurement to determine the amount of organic vapor and its corresponding emission wavelength. Through this way, we tried to figure out the quantitative relationship between the amount of organic vapor and the emission. For example, for the transition from 4CN(THF) (**G4**) to 4CN(Hex) (**G2**) (Supplementary Figure S43), the initial single emission peak at 495 nm will change to double peaks at 450 and 560 nm upon fuming with *n*-hexane. However, the fluctuated ratio of the emission intensity of 450 and 560 nm at different fuming time makes it very difficult to give an exact quantitative analysis. This is reasonable considering that the high sensitivity of fluorescence detection technology. We found that even place the same crystalline sample in the same holder with only different positions or angles, the fluorescent intensity may be different. This greatly hinders the quantitative analysis of the relationship between vapor and luminescence.

We have also tried to analyze the systems associated with non-white-emission, such as the transition from 4CN(Non1) (**G3**) to 4CN(ET2) (**G1**) (Supplementary Figure S39). The initial emission peak at 480 nm will gradually blue-shifts to 450 nm upon fuming with a mixed vapor (dichloromethane/ethanol) (Figure S1a). However, due to the using of mixed fuming solvents, the ^1H NMR spectra indicated that the mainly adsorbed solvents was volatile dichloromethane at the initial stage (less than 24 h, Figure S1b). As the time moves to 36 h, ethanol becomes dominative. Obviously, the emission wavelength is not determined by the amount of a single solvent. Therefore, it is very difficult to figure out the exact quantitative relationship between the amount of organic vapor and the light emission.

Figure S1. (a) Normalized fluorescence spectra of 4CN(ET2), 4CN(Non1) crystals and 4CN(Non1) crystals fumed in mixed vapor (dichloromethane/ethanol) for 0.5 h, 4 h, 8 h, 16 h, 24 h and 36 h. (b) ^1H NMR spectra of 4CN(Non1) crystals fumed in mixed vapor (dichloromethane/ethanol) for 0.5 h, 4 h, 8 h, 16 h, 24 h and 36 h.

Comment 1: Why the crystals are obvious dual emitted after hexane or heptane inclusion, and how the guests affect the relaxation of host. The difference of relax behavior between crystals inclusion of hexane and THF or other guests.

Response: Thanks for the valuable comments. We are sorry for the unclear interpretation of the mechanism of relaxation process which affect your understanding on white-light emission to a certain extent. The flexible environment of the framework is derived from two factors: the soft molecular conformations and dynamic intermolecular π - π interactions constitutes the flexible skeleton; the flexibility derived from highly disorder long-chain alkanes further renders the benzonitrile peripheries in the channel more space for movement in the excited state. This flexible environment can thus induce the relaxation from intrinsic emission in the excited state to generate a longer-wavelength emission. The yellow emission exhibits longer lifetime than that of blue one, which indicates its possibility of relaxing from short-wavelength emission in the excited state. As suggested from reviewer 1, the time-resolved emission spectra (TRES) of the white-light emissive crystals was measured to provide much more direct evidence. Upon increasing the gating time, the

progressively weakened shorter-wavelength emission and intensified longer-wavelength emission confirms directly that the yellow emission should be relaxed from the blue emission.

Compared with the “soft” linear and long-chain alkanes (*n*-hexane, *n*-heptane, *n*-octane, *n*-decane and *n*-dodecane), a significant difference of THF and other solvents is their structural “rigidity”. This can also be confirmed by the crystal structures. For example, the alkanes in the frameworks is highly disorder and very difficult to be refined, while THF and other solvents in the non-white-light emitting crystals is ordered and can be well refined. The absent of flexible environment in the non-white-light emitting crystals prevents the relaxation process in the excited state to generate the longer-wavelength emission, which thus can only exhibit single emission peak. For better understanding the physical picture behind white-light emission, the whole part of this section has been rewritten:

“We further wonder to explore the mechanism of white-light emission. Considering that all the other non-white-light emitting crystals (**G1**, **G3** and **G4**) exhibit blue (450 nm) to blue-green (500 nm) emission, the shorter-wavelength emission band (450-470 nm) of the white-light emitting **G2** can be reasonably classified as the intrinsic emission (Supplementary Fig. S21). The remaining question is the origin of the yellow emission (550-570 nm). After heavily grinding, we found that the dual emission bands disappear accompany with the emerging of a new peak at 537 nm (Fig. 3d). The PXRD spectra indicated the ground sample was almost an amorphous state (Fig. 3d). The molecules in amorphous state are in a relatively disordered and flexible environment. The origin of the amorphous yellow emission was thus thought to be induced by a flexible environment. Inspired by this, we propose that the molecule in the framework is also located in a similar flexible environment that is derived from two factors: the soft molecular conformation and dynamic intermolecular interactions constitute the flexible skeleton; the flexibility derived from highly disorder long-chain alkanes in the crystals further renders the benzonitrile peripheries in the channel more space for movement in the excited state. This flexible environment can thus allow relaxation in the excited state to generate a longer-wavelength emission.

To further verify our speculation, the photoluminescence decay curves of all the white-light emissive crystals were firstly measured. Their lifetime profiles were very similar whose dual emission were all evaluated to be nanosecond scale. This indicates both the blue and yellow emission can be ascribed to fluorescence (Supplementary Fig. S23). Noteworthy, the lifetimes of yellow emission peaks are about three or four times longer than those of blue emission peaks. For

example, for 4CN(Hex) crystal, its lifetimes are 1.2 ns for 452 nm and 5.1 ns for 560 nm, respectively (Supplementary Fig. S23a). This confirms that the longer-wavelength emission may originate from the relaxation of the short-wavelength emission in the excited state. More direct evidence was provided by time-resolved emission spectra (TRES) of the white-light emissive crystals. Taken 4CN(Hex) crystal as an example, it is obvious that the earlier time gate spectrum (0-4 ns) is characterized by blue emission with no contribution from yellow region, confirms the intrinsic character of blue emission (Figure 3e). With the time gate moves to prolonged time (4-7 ns), the PL spectrum exhibits a broad emission with two peaks located at 460 and 550 nm. However, the contribution from yellow emission becomes dominative during 7-100 ns time gate. The progressively weakened shorter-wavelength emission and intensified longer-wavelength emission confirms that the yellow emission should be relaxed from the blue emission. We can thus conclude, for the white-light with dual emission, its blue emission peak comes from the intrinsic emission of the framework whereas the yellow one should derived from the flexible solvent and dynamic framework-induced excited state relaxation (Fig. 3f).”

Comment 2: Maybe the author should further illustrate why the stronger π - π interaction result in the emission redshift.

Response: Thanks for the valuable comments. The stronger π - π interaction can cause effective frontier orbital interactions of a pair of molecules. For instance, the major electronic interactions involve their highest-energy filled (HO) and lowest-energy unfilled (LU) orbitals. According to the rules of perturbation theory, their HO will interact with each other to yield new HOs. Similarly, the LU will interact with each other to produce two new LUs. The new HOs and LUs are split in energy relative to the original ones. As a result, the stabilized HO is higher in energy than the original HO and the stabilized LU is lower in energy than the original LU. Thus, the energy gap between the new HO and LU will be lower and thus redshift the emission, caused by π - π interaction similar to the excimer case. For better understanding the emission mechanism of this four groups, the whole part of this section has been reorganized:

“The λ_{em} and θ of these crystals are shown in Supplementary Figure S7 and Table S1. The λ_{em} of the four groups generally conforms to the following sequence: **G1 < G2 < G3 < G4**. **The positive correlation between λ_{em} and θ is anticipated as that the large θ leads the poor conjugation and thus**

blueshifts the λ_{em} . On the other hand, the cofacial π - π interaction will redshift the λ_{em} as the interaction can stabilize the excited states. For **G1** and **G2**, the blue emission ranges from 440 to 470 nm can be attribute to the large dihedral angles ($\theta_{aver} \sim 55^\circ$). The θ_{aver} in **G4** are around 45° , much smaller than **G1** and **G2**, accounts for their good π -conjugation and relative red-shifted emission to 480-500 nm. A special case is **G3**, though the biggest dihedral angles ($\theta_{aver} > 70^\circ$) will destroy the molecular π -conjugation which should blue-shifted the emission, the strong intermolecular π - π interactions will force this trend to go in the opposite direction and finally leads to slightly red shift emission to cyan color (470-480 nm) (Supplementary Fig. S11c, d). In short, the colorful emission of crystallized 4CN can be ascribed to the synergy of flexible molecular conformation and dynamics intermolecular interactions.”

***Comment 3:** Emission enhancement and red-shifting of sample by adding n-hexane to acetone may not illustrate the flexibility of molecule. Various molecular conformation is fixed as the rapid aggregation process and form nonporous solid.*

Response: Thank you very much for pointing out this mistake. The description in the main text and experiment in the supplementary information related to 99.9% aggregate is deleted.

***Comment 4:** The author claims “porous molecular crystals”, but the porosity was not established by gas or vapor adsorption isotherm.*

Response: Thanks for the valuable comments. As we have mentioned in “General response to all the three reviewers” at the very beginning, the porous structure in this manuscript is not robust. PXRD profile of 4CN(Hex) crystals after gas adsorption-desorption is consistent with that of simulated nonporous 4CN(Non1) (Figure S24). This confirms that the porous crystals can transform into nonporous crystalline state during degassing. This non-robust character not only brings about multiple and switchable structural transformation between the 13 crystals, but also facilitates the relaxation process in the excited state which finally leads to white-light emission.

PMCs focuses much more on robustness, which may not be the most suitable name for our materials. In our manuscript, we want to emphasize the important role of channels, especially for the generation of white-light emission. Meanwhile, considering that the main intermolecular

interactions in the crystals is hydrogen bonds and the definition of hydrogen-bonded organic frameworks does not pose requirement on the robustness of the pore when the solvent is removed, which fits more closely with the flexible character of our materials. So, to more accurately describe our materials, the term “porous molecular crystals” was replaced by “hydrogen-bonded organic frameworks” in our manuscript. The description related to gas adsorption isotherm was added in the main text of the revised manuscript: “The gas adsorption-desorption experiments further confirmed the flexibility of the framework. After degassing at 110 °C under high vacuum for 12 h, the sample showed low uptake capacity for N₂ with BET surface area of 3.0 m²g⁻¹ at 77 K and 1 bar (Supplementary Fig. S24a). PXRD profile of 4CN(Hex) after gas adsorption-desorption was also consistent with that of simulated nonporous 4CN(Non1) (Figure S24b). This confirms that the kinetically stable porous crystals can transform into thermodynamically stable nonporous crystalline state.”

Supplementary Figure S24. (a) Sorption isotherms measured on 4CN(Hex) for N₂ at 77 K. (b) The PXRD spectra of simulated 4CN(Hex), 4CN(Non1) and 4CN(Hex) after gas adsorption-desorption.

Comment 5: *It is not clear that how stable and reversible is the skeleton. How many cycles can the crystals perform reversible light emission?*

Response: Thanks for the valuable comments. The stability and reversibility of the frameworks has been confirmed by the negligible fluorescence changes after multiple reversible transformations (Supplementary Figures S39, S41, S42 and S43). The relevant description was added in the main text of the revised manuscript: “The fluorescence shown no signs of fatigue under modest cycling demonstrated that these transitions were reversible (Supplementary Fig. S39b, S41b, S42b and S43b).”

Supplementary Figure S39. (a) Normalized fluorescence spectra and (b) fatigue resistance of reversible conversion between 4CN(ET2) (G1) and 4CN(Non1) (G3).

Supplementary Figure S41. (a) Normalized fluorescence spectra and (b) fatigue resistance of reversible conversion between 4CN(THF) (**G4**) and 4CN(Non1) (**G3**).

Supplementary Figure S42. (a) Normalized fluorescence spectra and (b) fatigue resistance of reversible conversion between 4CN(THF) (**G4**) and 4CN(Hex) (**G2**).

Supplementary Figure S43. (a) Normalized fluorescence spectra and (b) fatigue resistance of reversible conversion between 4CN(THF) (G4) and 4CN(ET2) (G1).

REVIEWER COMMENTS

Reviewer #1 (Remarks to the Author):

The authors have answered all the queries of the reviewers quite satisfactorily with new experimental data. A drastic modification in the write-up was also carried out, including title, results and discussion, experimental sections, and references. The clarity and quality of the revised manuscript are satisfactory to me, and it is likely to cater to the interest of the wider scientific community. Hence, the publication of the manuscript is recommended in its present form.

Reviewer #2 (Remarks to the Author):

The description in this paper contains careful answers to several questions I had previously raised. I have found all of the answers to be satisfactory. Therefore, I recommend that this paper be accepted in its present form.

Reviewer #3 (Remarks to the Author):

The authors have done great job in revising the manuscript, and my previous concerns have been fully addressed. Therefore, I would like to recommend it for publication in Nature Communication as it is. Congratulations!

Responses to reviewers' comments for the manuscript titled **“Multiple yet switchable hydrogen-bonded organic frameworks with white-light emission”**

Reviewer #1 (Remarks to the Author):

The authors have answered all the queries of the reviewers quite satisfactorily with new experimental data. A drastic modification in the write-up was also carried out, including title, results and discussion, experimental sections, and references. The clarity and quality of the revised manuscript are satisfactory to me, and it is likely to cater to the interest of the wider scientific community. Hence, the publication of the manuscript is recommended in its present form.

Responses: We sincerely thank the reviewer's recognition and support of publication of our work.

Reviewer #2 (Remarks to the Author):

The description in this paper contains careful answers to several questions I had previously raised. I have found all of the answers to be satisfactory. Therefore, I recommend that this paper be accepted in its present form.

Responses: We sincerely thank the reviewer for the comments and supporting publication of our work.

Reviewer #3 (Remarks to the Author):

The authors have done great job in revising the manuscript, and my previous concerns have been fully addressed. Therefore, I would like to recommend it for publication in Nature Communication as it is. Congratulations!

Responses: We would like to thank you very much for your recognition of our work and valuable comments.